# BANDIT LEARNING IN MATCHING: UNKNOWN PREFERENCES ON BOTH SIDES

## ABSTRACT

Two-sided matching under uncertainty has recently drawn much attention due to its wide applications. Matching bandits model the learning process in matching markets with the multi-player multi-armed bandit framework, i.e. participants learn their preferences from the stochastic rewards after being matched. Existing works in matching bandits mainly focus on the one-sided setting (i.e. arms are aware of their own preferences accurately) and design algorithms with the objective of converging to stable matching with low regret. In this paper, we consider the more general two-sided setting, i.e. participants on both sides have to learn their preferences over the other side through repeated interactions. Specifically, we formally introduce the two-sided setting and consider the rational and general case where arms adopt "sample efficient" strategies. Facing the challenge of unstable and unreliable feedback from arms, we design an effective algorithm that requires no restrictive assumptions such as special preference structure and observation of winning players. Moreover, our algorithm is the first to provide a theoretical upper bound and achieves $O(\log T)$ regret which is proved optimal in terms of $T$.

## 1 INTRODUCTION

Stable matching with preferences on both sides is a classic problem with wide applications encompassing marriage, college admission and labor markets. The classical literature Roth & Sotomayor (1992); Roth & Xing (1997); Gale & Shapley (1962) that studies matching markets always assumes that every participant is aware of her own preference perfectly beforehand, which may not be satisfied in many scenarios. Under such uncertainty, a crucial question of matching markets is the convergence to equilibrium. In online marketplaces (e.g. Upwork, TaskRabbit), repeated decentralized matching between the demand side and the supply side becomes a core process. To characterize such process, Liu et al. (2020) introduces the model of matching bandits which adopts the multi-armed bandit (MAB) model in matching markets. MAB is a classic and well-studied framework that models the decision-making process under uncertainty(Katehakis & Veinott Jr (1987); Auer et al. (2002)). An player faces $K$ arms with different utilities and aims to find out the best arm based on the stochastic reward received after each pull. When studying matching markets with the MAB framework, not only the regret, but also the stability should be taken account into consideration.

A strand of literature (e.g., Liu et al. (2020); Basu et al. (2021); Liu et al. (2021); Sankararaman et al. (2021); Maheshwari et al. (2022)) proposes algorithms with the objective of achieving stable matching with low regret. However, one limitation is that existing models of matching bandits all assume perfect knowledge of preference on one side's (i.e. each arm knows its own preference). We refer this setting as the one-sided setting. Arms are able to give precise and stable feedback all the time in the one-sided setting.

In our work, we study the more general and challenging case where even arms lack the knowledge of arm preferences, i.e. the two-sided setting. As the players' decisions are based on arms' feedback, it is crucial that arms can efficiently learn their preferences so as to return useful feedback to players. Therefore, the learning speed of arms is the key to the problem. In this paper, we measure the learning difficulty of the arm side by comparing it with the player side and we consider the reasonable case where the difficulty level of arms' preferences learning is comparable with players' up to a constant $D$. As arms are also uncertain about their preferences and need to learn through interactions, arms will keep track of the rewards received after every match associated with each player and choose to

match with a player based on historical payoffs. Inspired by two mostly used strategies: arms (1) intuitively sort the players by empirical means, or (2) use the UCB method which is proved to obtain sub-linear regret, we propose the definition of "sample efficient" to characterize the learning efficiency of arms' strategies and further analyze the more general case where arms' strategies are "sample efficient". Due to the generality, we consider the fully decentralized setting without restrictions, i.e. no central communicator, no external available information such as observation and no assumption on the preference structures. We propose our algorithm: Round-Robin ETC in this two-sided setting and provide rigorous regret analysis.

## 1.1 CHALLENGES AND CONTRIBUTIONS

We introduce the more general case of matching bandits, i.e. two-sided setting and further specify the model of two-sided matching bandits. In particular, we formally model the two-sided setting, consider arms' strategies and define the new notion "sample efficiency" to characterize arms' learning efficiency.

We propose a new algorithm for the complex two-sided setting and provide rigorous regret analysis. In the two-sided setting, arms may give incorrect feedback, invaliding many commonly used techniques in the one-sided setting, and making stable equilibrium more challenging to achieve. To the best of our knowledge, we are the first to propose algorithms with provable upper bounds of regret based on this setting in matching bandits. Moreover, we provide new techniques in analysing the matching bandits.

We relax several common assumptions while achieving near optimal regret simultaneously. When considering matching markets, previous work usually introduce strong assumptions on the structure of preferences, such as globally ranked players in Sankararaman et al. (2021) and $\alpha$-reducibility in Maheshwari et al. (2022). Observation of winning players is also a strong but commonly used assumption in matching bandits (Kong & Li (2023); Liu et al. (2021); Kong et al. (2022)). Our proposed algorithm Round Robin ETC can be applied in both the one-sided setting and two-sided setting with minor modification, and there is no such strong assumptions required for our algorithm. Moreover, our algorithm achieves $O(\log T/\Delta^2)$ regret with respect to player-optimal stable matching, where $T$ represents the time horizon and $\Delta$ represents the minimal gap of arm utilities. This regret bound is tight in terms of $T$ and $\Delta$. The regret bound also matches with the state-of-art result in the simpler one-sided setting.

## 1.2 RELATED WORK

Table 1. Comparison between our work and prior results.

|  | Assumptions | Player-Stable Regret |
|---|---|---|
| Liu et al. (2020) | one-sided, centralized, known $\Delta$ | $O(K \log T/\Delta^2)*$ |
| Liu et al. (2020) | one-sided, centralized | $O(NK^3 \log T/\Delta^2)$ |
| Sankararaman et al. (2021) | one-sided, globally ranked | $O(NK \log T/\Delta^2)$ |
| Basu et al. (2021) | one-sided, uniqueness consistency | $O(NK \log T/\Delta^2)$ |
| Maheshwari et al. (2022) | one-sided, $\alpha$-reducibility | $O(CNK \log T/\Delta^2)$ |
| Liu et al. (2021) | one-sided, observation | $O(\exp{(N^4)}K^2 \log^2 T/\Delta^2)$ |
| Kong et al. (2022) | one-sided, observation | $O(\exp{(N^4)}K^2 \log^2 T/\Delta^2)$ |
| Basu et al. (2021) | one-sided | $O(K \log^{1+\epsilon} T/\Delta^2 + \exp{(1/\Delta^2)})*$ |
| Kong & Li (2023) | one-sided, observation | $O(K \log T/\Delta^2)*$ |
| Zhang et al. (2022) | one-sided | $O(NK \log T/\Delta^2)*$ |
| this paper | two-sided | $O(K \log T/\Delta^2)*$ |

[1] $K$ is the number of arms and $N$ represents the number of players.
[2] $*$ represents the type of regret bound is player-optimal stable regret.
[3] $C$ is related to the preference structure and it may grows exponentially in $N$.
[4] $\epsilon$ is a positive hyper-parameter.

The first work that combines MAB framework and matching market is from Das & Kamenica (2005), and Das & Kamenica (2005) propose algorithm with numerical study under the strong assumption of sex-wide homogeneity. Liu et al. (2020) generalize the MAB based matching and propose basic ETC type and UCB type algorithms. However, Liu et al. (2020) mainly consider the centralized setting which is not so practical in reality.

Later, a strand of works that studies the one-sided decentralized matching bandits emerges. As we stated before, there are many works that make different assumptions on arm preferences. Sankararaman et al. (2021) analyse the setting of globally ranked players where all the arms sort players the same. Later, Basu et al. (2021) consider a more general case of uniqueness consistency and propose UCB-D4. Another special case of $\alpha-$reducibility is studied by Maheshwari et al. (2022). These assumptions all ensure one unique stable matching. Without restriction on preferences, it is common that the market could have multiple stable matches. Denote the least preferred stable matching for players by player-pessimal stable matching and the most preferred one by player-optimal stable matching. Regret defined on the optimal stable matching is more desired as comparing with the pessimal stable matching could induce an extra linear regret than the optimal stable matching. With accurate knowledge of arm preferences on both arm side and player side, Liu et al. (2021) design a conflict avoiding algorithm named CA-UCB which upper bound the player-pessimal stable regret under the assumption of "observation". Similarly, TS based conflict avoiding algorithm with "observation" is analysed by Kong et al. (2022). Basu et al. (2021) propose a phased-based algorithm but with a high exponential dependency on $\frac{1}{\Delta}$. Adopting the assumption of "observation", Kong & Li (2023) propose ETGS which guarantees player-optimal stable regret. ML-ETC proposed by Zhang et al. (2022) is a ETC based algorithm that can apply to general preference structures, and it also upper bounds the player-optimal stable regret without "observation".

The above literature mostly requires knowledge of arm preferences and rely on the precise feedback of arms. The work from Pokharel & Das (2023) considers the case of both side unknown preferences in matching bandits. Pokharel & Das (2023) propose PCA-DAA using the random delay to reduce the likelihood of conflicts, but only empirical results are provided. Some works study two-sided matching bandits from other aspects. For instance, Jagadeesan et al. (2023) investigate matching markets under the stochastic contextual bandit model, where, at each round, the platform selects a market outcome with the aim of minimizing cumulative instability.

## 2 SETUP

Suppose there are $N$ players and $K$ arms, and denote the set of players and arms by $\mathcal{N}$ and $\mathcal{K}$ respectively. We adopt the commonly used assumption in matching bandits that $N \leq K$ ( e.g. Liu et al. (2021); Kong & Li (2023); Basu et al. (2021); Liu et al. (2020); Basu et al. (2021)). Both the player side and arm side are unaware of their preferences. Specifically, for each player $j$, she has a fixed but unknown utility $u_{jk}$ associated with each arm $k$ and prefers arm with higher utility. For each arm $k$, it also has a fixed but unknown utility $u_{kj}^a$ associated with each player $j$ and prefers player with higher utility (the superscript $a$ stands for "arm"). Without loss of generality, we assume all utilities are within $[0, 1]$, i.e. for every $j \in \mathcal{N}, k \in \mathcal{K}, u_{jk}, u_{kj}^a \in [0, 1]$. Define the utility gap for player $j$ as $\Delta_j = \min_{k_1, k_2 \in \mathcal{K}, k_1 \neq k_2} |u_{jk_1} - u_{jk_2}|$ and the utility gap for arm $k$ as $\Delta_k^a = \min_{j_1, j_2 \in \mathcal{N}, j_1 \neq j_2} |u_{kj_1}^a - u_{kj_2}^a|$. As a common assumption in previous work (e.g. Pokharel & Das (2023); Liu et al. (2020; 2021), all preferences are strict, which means that both the minimal gap of player $\Delta = \min_{j \in \mathcal{N}} \Delta_j$ and the minimal gap of arm $\Delta^a = \min_{k \in \mathcal{K}} \Delta_k^a$ are positive. Moreover, we consider the reasonable case where the difficulty level of arms' preferences learning is comparable with players' up to a positive constant $D \in (0, \infty)$. Specifically, we assume $D\Delta^a \geq \Delta_j$ in this paper. Throughout the time horizon $T$, every player and arm will learn about their own preferences through interactions and want to match with one from the other side with higher utility. We use the notation $j_1 \succ_k j_2$ to indicate that arm $k$ prefers player $j_1$ than player $j_2$ and the similar notation $k_1 \succ_j^a k_2$ to represent that player $j$ prefers arm $k_1$ than arm $k_2$.

At each time step $t \leq T$, each player $j$ pulls an arm $I_j(t)$ simultaneously. If there exists one player pulling the arm $k$, we assume that the arm $k$ will choose to match with the player rather than staying unmatched since all utilities are non-negative. When there are multiple players pulling arm $k$, a conflict arise, and arm $k$ will choose to match one of the candidates based on its strategy (see details in Section 2.1). The unchosen players will get rejected and obtain no reward. Denote the winning

player on arm $k$ at time step $t$ by $A_k(t)$. Let $C_j(t)$ represent the rejection indicator of player $j$ at time step $t$. $C_j(t) = 1$ indicates that player $j$ gets rejected and $C_j(t) = 0$ otherwise. When a match succeeds between player $j$ and arm $k$, both player $j$ and arm $k$ receive stochastic rewards sampled from the fixed latent 1-subgaussian distributions with mean $u_{jk}$ and $u_{kj}^a$, respectively. In this paper, we consider the general decentralized setting, i.e., no direct communication among players is allowed, and there is no central organizer or extra external information such as observation.

**No Observation of Winning Players.** In the literature of matching bandits, observation of winning players (which assume that all players can observe the winning player on every arm) is a strong but widely used assumption. Even when some arms are not selected by the player, the player can also get their information based on observation. This assumption greatly helps players to learn arms' preferences and other players' actions. Liu et al. (2021) incorporate the observation to design a conflict-avoid algorithm, Kong & Li (2023) use the observation to help players infer others' learning progress easily. However, it will be more challenging but more desirable to throw away the assumption. In real applications, the common case is that a player will only be informed her own result (success or rejection) rather than being aware of every accepted player for every company. The assumption of no observation also captures the fully decentralized scenario, i.e. players take actions only based on their own matching histories, without access to others' information.

## 2.1 ARMS' STATEGIES

At each time step $t$, if there exist multiple players pulling the arm $k$, arm $k$ will choose to match one player according to past rewards received. Instead of considering a specific strategy, we consider the general case where arms can adopt different strategies as long as they can efficiently learn their own preferences. The empirical mean associated with player $j$ estimated by arm $k$ is denoted by $\hat{u}_{kj}^a$ and the matched times associated with player $j$ estimated by arm $k$ is denoted by $N_{kj}^a$. Define event $\mathcal{E}^a = \{\forall j \in \mathcal{N}, k \in \mathcal{K}, |\hat{u}_{kj}^a - u_{kj}^a| < 2\sqrt{\frac{\log T}{N_{kj}^a}}\}$. $\mathcal{E}^a$ represents that the samples' quality is not too bad so that the empirical means are very far from real values at every time slot. We will show in our proof that $\mathcal{E}^a$ is a high-probability event since all samples are drawn from sub-gaussian distributions.

**Definition 1** (Arms' Sample Efficiency). *We say arm $k$ adopt $R$ sample efficient strategy, if after collecting $R\frac{\log T}{(\Delta^a)^2}$ samples for every player, conditional on $\mathcal{E}^a$, arm $k$ will choose to match with the player with highest utility among the candidates.*

Sample efficient strategies enable arms to provide useful feedback soon. If arms adopt sample efficient strategies, they will not choose sub-optimal candidates a lot as long as the samples are not so bad. Several commonly used bandit-learning methods like UCB policy and following the empirical leader satisfies the sample efficiency. Since arms will also receive rewards and prefer players with higher utilities, it is reasonable for rational arms to adopt efficient learning strategies. Thus, in this paper, we consider the case where arms adopt sample efficient strategies.

## 2.2 REGRET FOR BOTH SIDES

Before we introduce the definition of regret, we recall the definition of matching stability, which is an important issue when considering matching bandits in matching markets.

A matching between player side and arm side is stable if there does not exist two (arm, player) matches such that each one prefers the other partner than the current matched partner. Let $m_j$ represent the matched pair of player $j$ in the matching $m$. For each player $j$, her optimal stable arm $\overline{m}_j$ is the arm with highest utility among her matched arms in all possible stable matchings while her pessimal stable arm $\underline{m}_j$ being the matched arm with lowest utility. For each player $k$, its optimal stable player $\overline{m}_k^a$ is the player with highest utility among its matched players in all possible stable matchings while its pessimal stable player $\underline{m}_k^a$ being the matched player with lowest utility. Unlike previous work, we consider the stable regret for both the player side and the player side. The player-optimal and player-pessimal stable regret for player $j$ are defined as follows, respectively:

$$\overline{R}_j(T) = \mathbb{E}[\sum_{t=1}^{T}(u_{j\overline{m}_j} - (1 - C_j(t))u_{jI_j(t)})].$$

$$\underline{R}_j(T) = \mathbb{E}[\sum_{t=1}^{T}(u_{j\underline{m}_j} - (1 - C_j(t))u_{jI_j(t)})].$$

Similarly, the arm-optimal and arm-pessimal stable regret for arm $k$ are defined as follows, respectively:

$$\overline{R}_k^a(T) = \mathbb{E}[\sum_{t=1}^{T}(u_{k\overline{m}_k^a}^a - u_{kA_k(t)}^a)].$$

$$\underline{R}_k^a(T) = \mathbb{E}[\sum_{t=1}^{T}(u_{k\underline{m}_k^a}^a - u_{kA_k(t)}^a)].$$

**Player-optimal Stable Regret.** Note that the optimal stable regret is defined with respect to the optimal stable pair that has higher utility than the pessimal stable pair. Thus, it is more challenging and desired to achieve sublinear optimal stable regret. However, as a result in Gale & Shapley (1962), it is impossible to get both sublinear player-optimal stable regret and sub-linear arm-optimal stable regret. Note that Gale & Shapley (1962) also propose the Gale-Shapley (GS) algorithm which obtains optimal stable matching for the proposing side. We wonder if the similar result holds in the two-sided unknown setting. Thus, in this paper, we mainly focus on player-side strategies and player-optimal stable matching.

## 3 ROUND-ROBIN ETC ALGORITHM

In this section, we propose our algorithm: Round-Robin ETC which obtains an asymptotic $O(\log T)$ player-optimal stable regret.

### 3.1 BRIEF INTRODUCTION

In this subsection, we will discuss some unique challenges in the two-sided matching bandits, as well as how our proposed techniques address the challenges. Then, we give a brief introduction of the major phases in our algorithm.

In learning problems, trade-off between exploration and exploitation is usually a key issue. As for matching bandits, players need to consider whether to explore in order to learn preferences or to exploit by starting the player-optimal stable matching process based on current estimations.

The exploration is more complicated when studying matching bandits. Unlike the traditional MAB problem, in matching bandits, only when an player does not get rejected, can she receive reward to learn preferences. Moreover, when considering the two-sided setting, arms also need to learn their utilities through rewards received after each match. Since the convergence to the optimal stable matching requires both the player-side and the arm-side to have accurate estimations. It is essential for both players and arms to get enough samples. Furthermore, a unique challenge brought by the two sided setting lies in the asymmetry of the learning ability on both sides. Intuitively, it will be harder for arms to collect enough samples since players can choose arms proactively while arms can only passively choose one player from the candidates. But it is crucial for arms to learn their preferences correctly early so that players can receive true information when conflicts happen. Notice that when there is no conflict, i.e. only one player that pulls one arm, the arm and the player will get matched. Such successful match generates a clear sample for both the player and the arm. We use the idea of round-robin exploration to avoid conflict and enable both the player-side and the arm-side to learn their preferences simultaneously.

The exploitation also becomes different and more challenging when considering matching bandits. The exploitation in matching bandits is to form optimal stable matchings among players and arms. Reaching the correct optimal stable matching requires cooperation between all players and arms given limited communication. It is crucial to let players decide on when to end their individual exploration and to start a collective matching process. As players and arms don't have explicit communication channel or direct observation of conflict results, players can hardly learn other players' exploration progress. It is even harder for players to learn arms' exploration progress since players can only infer that from arms' passive actions ( i.e. choosing one player from the candidates). To address these

challenges, we incorporate the confidence bounds to enable players to measure their own exploration progress and wait for arms to get enough samples simultaneously. We also design decentralized communication through deliberate conflicts, which allow players to send and infer information. Specifically, players will deliberately compete for an arm, trying to send information by letting other player get rejected or to receive information by inferring from the rejection indicators. Furthermore, we carefully design the algorithm such that players can enter exploitation as soon as possible, i.e., they do not need to wait until all others have learned their preferences accurately. The intuitive idea is that, if a player is to start exploitation, she only needs to make sure that any other player that could potentially "squeeze" her out has already entered (or also about to enter) exploitation.

Together with these analysis, we provide the brief introduction of our algorithm. Firstly, the algorithm will assign distinct index to each player. Next, players will do rounds of round-robin exploration. After every round of exploration, players will communicate their progress of preference learning to decide on whether to start matching. If players decide to start matching, they will run the Gale-Shapley (GS) algorithm and occupy their potential optimal stable arm till the end. Otherwise, the players will start a new round of exploration.

## 3.2 ROUND-ROBIN ETC

---

**Algorithm 1** Round Robin ETC (for an player $j$)

---
    # Phase 1: Index Assignment
  1: Index $\leftarrow$ *INDEX-ASSIGNMENT(N, $\mathcal{K}$)*
    # Phase 2: Round Robin
  2: $N_2 \leftarrow N, \mathcal{K}_2 \leftarrow \mathcal{K}, K_2 \leftarrow K$
    # $N_2$ denotes the number of remaining players in Phase 2, $\mathcal{K}_2$ denotes the remaining arms
  3: **while** OPT$= \emptyset$ **do** # when $j$ hasn't found her potential optimal stable arm yet
    #Sub-Phase: Exploration
  4:     (Success, $\hat{\boldsymbol{u}}_j, \boldsymbol{N}_j) \leftarrow$ *EXPLORATION*(Index, $K, K_2, \mathcal{K}_2, \hat{\boldsymbol{u}}_j, \boldsymbol{N}_j)$
    #Sub-Phase: Communication
  5:     Success $\leftarrow$ *COMM*(Index, Success, $N_2, K_2, \mathcal{K}_2)$
    #Sub-Phase: Update
  6:     OPT $\leftarrow$ *GALE-SHAPLEY*(Success, $N_2, \mathcal{K}_2, \hat{\boldsymbol{u}}_j, \boldsymbol{N}_j)$
  7:     $N_1 \leftarrow N_2, \mathcal{K}_1 \leftarrow \mathcal{K}_2$
  8:     **if** Success$= 1$ **then Break while**#successful players will enter the exploitation phase
  9:     **end if**
10:     **for** $t = 1, ..., N_2 K_2$ **do**
11:         **if** $t = ($Index $- 1)K_2 + m$ **then** # check arms' availability
12:             Pull arm $k$ that is $m$-th arm in $\mathcal{K}_2$
13:             **if** $C_j = 1$ **then** $\mathcal{K}_1 \leftarrow \mathcal{K}_1 \setminus \{k\}, N_1 = N_1 - 1$
14:             **end if**
15:         **end if**
16:     **end for**
17:     $N_2 \leftarrow N_1, \mathcal{K}_2 \leftarrow \mathcal{K}_1$ #update available arms and number of players
18:     Index $\leftarrow$ *INDEX-ASSIGNMENT($N_2, \mathcal{K}_2$)*
19: **end while**
    #Phase 3: Exploitation Phase:
20: Pull OPT arm

    **procedure** *INDEX-ASSIGNMENT*$(N, \mathcal{K})$
  1: $\pi \leftarrow \mathcal{K}[1]$
  2: **for** $t = 1, 2, ..., N$ **do**
  3:     Pull arm $\pi$
  4:     **if** $C_j = 0, \pi = \mathcal{K}[1]$ **then**
  5:         Index $\leftarrow t, \pi \leftarrow \mathcal{K}[2]$
  6:     **end if**
  7: **end for**
  8: **return** Index

---

The Algorithm 1 consists of 3 phases: "Index Assignment", "Round Robin" and "Exploitation". Players will enter "Index Assignment" phase and "Round Robin" phase simultaneously, but may leave the "Round-Robin" phase for the "Exploitation" phase at different time steps.

In the "Index Assignment" phase (Line 1), every player will receive a distinct index. To be specific (see procedure *INDEX-ASSIGNMENT*), every player will keep pulling arm 1 until the first time, say step $t$, she doesn't get rejected. She will be assigned index $t$ and then move to pull the next arm, i.e., arm 2. Since there can only be one player that successfully matches with arm 1 at each time step, after $N$ time steps, all players can receive different indices.

---

**Algorithm 2** *EXPLORATION* (for player $j$)

---

**Require:** Index, $K_1, K, \mathcal{K}, \hat{\boldsymbol{u}}_j, \boldsymbol{N}_j$
 1: **for** $t = 1, 2, ..., K K_1^2 \lceil \log T \rceil$ **do**
 2:     Pull (Index $+ t$) $\mod K = m$-th arm in $\mathcal{K}$ and update $\hat{u}_{jk}, N_{jk}$
 3: **end for**
 4: **if** for every $k_1 \neq k_2 \in \mathcal{K}$, $\text{UCB}_{jk_1} < \text{LCB}_{jk_2}$ or $\text{LCB}_{jk_1} > \text{UCB}_{jk_2}$ **then**
 5:     Success $\leftarrow 1$ # the player achieves a confident estimation
 6: **end if**
 7: **return** Success, $\hat{\boldsymbol{u}}_j, \boldsymbol{N}_j$

---

---

**Algorithm 3** *COMM* (for player $j$)

---

**Require:** Index, Success, $N, K, \mathcal{K}$
 1: **for** $i = 1, 2, ..., N$,t_index $= 1, 2, ..., N$, r_index $= 1, 2, ..., N$, r_index$\neq$t_index **do**
     # player with Index=t_index is the transmitter and player with Index=r_index is the receiver
 2:     **for** $m = 1, 2, ..., K$ **do** # the communication process is through conflict on the $m$-th arm
 3:         **if** Index=t_index **then** # if transmitter
 4:             **if** Success$= 0$ **then** Pull the $m$-th arm in $\mathcal{K}$
 5:             **end if**
 6:         **end if**
 7:         **if** Index=r_index **then** # if receiver
 8:             Pull the $m$-th arm in $\mathcal{K}$
 9:             **if** $C_j = 1$ **then** Success$= 0$
10:             **end if**
11:         **end if**
12:     **end for**
13: **end for**
14: **return** Success

---

In the "Round Robin" phase (Line 3-19), the players will explore the arms without conflict, communicate on their progress of exploration, and update their indices and available arms in a round based way.

Each round will consist 3 sub-phases: exploration, communication and update. An player will leave the "Round Robin" phase when she finds out her optimal stable arm confidently. Then, she will enter the "Exploitation" phase and occupy her potential optimal stable arm, say arm $k$, making arm $k$ unavailable to other players. Denote the set of players that are still in the "Round Robin" phase by $\mathcal{N}_2$, the number of remaining players by $N_2$, the available set of arms by $\mathcal{K}_2$, and the number of available arms by $K_2$. We further elaborate on the three sub-phases in "Round Robin" below.

1. Exploration (Line 4, see Algorithm 2 *EXPLORATION*). Every player will explore available arms according to the index to avoid conflict, and every exploration will lasts for $K_2 K^2 \lceil \log T \rceil$ time steps. Based on the distinct index and the assumption that $K \geq N$, there will be at most one single player that pulls each arm at each time step during the exploration. Player $j$ will update her empirical mean $\hat{\boldsymbol{u}}_j$ and the matched times $\boldsymbol{N}_j$ throughout the exploration. The notions of upper confidence bound "UCB" and lower confidence bound "LCB" are defined as follows:

$$\text{UCB}_{jk} = \hat{u}_{jk} + c\sqrt{\frac{\log T}{N_{jk}}}, \text{LCB}_{jk} = \hat{u}_{jk} - c\sqrt{\frac{\log T}{N_{jk}}}, \tag{1}$$

where $\hat{u}_{jk}$ denotes the empirical mean and $N_{jk}$ denotes the times player $j$ is matched with arm $k$. Let $c = \max\{2, \frac{\sqrt{R}D}{2} + 1\}$. We say that when an player $j$ achieves a confident estimation on the arm set $\mathcal{K}^*$ if for every $k_1, k_2 \in \mathcal{K}^*$ such that $k_1 \neq k_2$, either $\text{UCB}_{jk_1} < \text{LCB}_{jk_2}$ or $\text{LCB}_{jk_1} > \text{UCB}_{jk_2}$ holds.

2. Communication (Line 5, see Algorithm 3 *COMM*). The players will communicate through deliberate conflicts in an index-based order. This sub-phase let players communicate on their progress of exploration. Specifically, they will communicate whether they have achieved confident estimations and the communication proceeds pairwise following the order of index. Player with index 1 will first serve as a transmitter, sending information to the player with index 2, then to player with index 3, 4 and so on. After player 1 have finished sending information to others, player 2 will be the transmitter, then player 3 and so on. The player who wants to receive information is the receiver.

   The communication subphase conducts all pairwise communication between all pairs of remaining players on all available arms for $N_2$ times. Specifically, for every pair of different remaining players $j_1$ and $j_2$, communication occurs on every available arm for $N_2$ times. Every communication is conducted through a deliberate conflict on the $m$-th arm of $\mathcal{K}_2$ between a transmitter and a receiver. The player with index "t_index", denoted as player $j_1$, serves as the transmitter, and the player with index "r_index" is the receiver. Suppose $j_2$ is the receiver, and arm $k$ is the $m$-th arm of $\mathcal{K}_2$. The receiver $j_2$ will choose arm $k$ to receive information. The transmitter $j_1$ will choose arm $k$ only when she fails to achieve a confident estimation or has been rejected when receiving others' information in the previous time steps during the communication sub-phase. Other players will pull an arbitrary arm $k' \neq k$.

   If a player achieves a confident estimation and never gets rejected when receiving others' information during the communication sub-phase, we say that the player obtains successful learning. Note that if a player obtains successful learning, it means that with high probability, the remaining players that may "squeeze" her out on the available arms all achieve confident estimations (and all obtain successful learning). We use "Success" in the pseudocode (Line 4, 5, 8) to denote the success signal, and "Success= 1" indicates that the player obtains successful learning while "Success= 0" otherwise. We call the players who obtain successful learning the successful players, and others are called unsuccessful players.

3. Update (Line 6-23). The successful players will be able to find out their potential optimal stable arms, and unsuccessful will update their indices, the number of remaining players $N_2$, and the set of available arm $\mathcal{K}_2$. The first procedure *GALE-SHAPLEY* (Gale & Shapley (1962)) enables successful players to match their potential optimal stable arms. Then successful players will enter the "Exploitation" phase, and unsuccessful players will update the available arms in order. Specifically, when $t = (n-1)K_2 + m$ in Line 11, the player with index $n$, suppose player $j$, will pull the $m$-th arm in $\mathcal{K}_2$, suppose arm $k$, to check its availability. If player $j$ gets rejected, then she will kick arm $k$ out of the available arm set. Lastly, unsuccessful players will update their indices by the *INDEX-ASSIGNMENT* function and start a new round.

In the "Exploitation" phase (Line 20), every player will keep pulling her potential optimal stable arm till the end.

### 3.3 REGRET ANALYSIS

**Theorem 1.** *If every player runs Algorithm 1, and arms adopt $R$ sample efficient strategies, then the optimal stable regret of any player $j$ can be upper bounded by :*

$$\overline{R}_j(T) \leq N + K^3 r \lceil \log T \rceil + Nr(KN(N-1) + N + K + 1) + 4KN + 2$$
$$= O(\frac{K \log T}{\Delta^2}) \tag{2}$$

*Moreover, the arm-pessimal stable regret for any arm $k$ can be upper bounded by:*

$$\underline{R}_k^a(T) \leq N + K^3 r \lceil \log T \rceil + Nr(KN(N-1) + N + K + 1) + 4KN + 2$$
$$= O(\frac{K \log T}{\Delta^2}) \tag{3}$$

*where $r$ equals to $\lceil \frac{4(c+2)^2}{K^2 \Delta^2} \rceil$ and $c = \max\{2, \frac{\sqrt{R}D}{2} + 1\}$.*

**Dependency on Parameters.** As a result in Sankararaman et al. (2021), the regret bound is optimal in terms of $T$ and $\Delta$. Compared with previous work, we share the same dependency on $K$ with the best existing algorithm that study the one-sided setting.

*Proof Sketch.* We only give a proof sketch for the player-optimal stable regret, the result for the arm regret can be obtained similarly.

Define the event $\mathcal{E} = \{\forall j \in \mathcal{N}, k \in \mathcal{K}, |\hat{u}_{jk} - u_{jk}| < 2\sqrt{\frac{\log T}{N_{jk}}}\}$. We can decompose the regret depending on whether $\mathcal{E}$ and $\mathcal{E}^a$ holds, i.e.

$$\overline{R}_j(t) = \mathbb{E}[\sum_{t=1}^{T}(u_{j\overline{m}_j} - (1 - C_j(t))u_{jI_j(t)})|\mathcal{E} \cap \mathcal{E}^a] + \mathbb{E}[\sum_{t=1}^{T}(u_{j\overline{m}_j} - (1 - C_j(t))u_{jI_j(t)})|\neg(\mathcal{E} \cap \mathcal{E}^a)]$$

$$\leq \mathbb{E}[\sum_{t=1}^{T}(u_{j\overline{m}_j} - (1 - C_j(t))u_{jI_j(t)})|\mathcal{E} \cap \mathcal{E}^a \neq] + TPr[\neg\mathcal{E}] + TPr[\neg\mathcal{E}^a]. \tag{5}$$

While the probability of $\neg\mathcal{E}$ and $\neg\mathcal{E}^a$ can be upper bounded by a $\frac{1}{T}$ factor, we only need to bound the regret conditional on $\mathcal{E} \cap \mathcal{E}^a$. By the design of the algorithm, we can easily find out that the initialization phase lasts for $N$ time steps, which means there will be at most $N$ regret caused by the initialization phase. As for the other two phases, we can prove the following statements:

- Conditional on $\mathcal{E}$ and $\mathcal{E}^a$, with probability more than $1 - \frac{2}{T}$, when a player achieves a confident estimation on the available arm set $\mathcal{K}_2$, the arms in $\mathcal{K}_2$ give accurate feedback.

- If arms in $\mathcal{K}_2$ give accurate feedback, the pulls of unsuccessful players will not influence the output of the potential optimal arms (i.e. *OPT* in Line 6) for successful players.

Then according to the design of the algorithm, the property of *GALE-SHAPLEY* and these statements, we can also prove that conditional on $\mathcal{E} \cap \mathcal{E}^a$, after no more than $O(\log T)$ time steps in the round-robin phase, all players will enter the exploitation phase with their correct optimal stable arm. Combining these all together, we can obtain the results.

□

## 4 CONCLUSION

In this work, we study the matching bandits in the two-sided setting where both the player side and arm side do not know their own preferences. We further model the details of two-sided setting, consider the general case with no restrictive assumptions, and propose Round-Robin ETC. In particular, we introduce the two-sided model formally and specify arms' strategies on how to choose players. We throw away the assumption of observations, make no restriction on the preference structure and study the general two-sided setting. Our algorithm obtains an $O(\log T)$ player-optimal stable regret and achieves the same order as the state-of-the-art guarantee in the simpler one-sided setting. To the best of our knowledge, we are the first to give theoretical results when considering the matching bandits with unknown preference on both sides.

**Future Direction.** In this paper, we mainly focus on the setting where arms adopt sample efficient strategies to choose players to match with. It remains to investigate the case where dishonest arms exist and may misreport their own preferences after they have already learned their preferences confidently. The future direction may focus on the case where arms also adopt more strategic strategies. The game playing between player-side and arm-side may also be an interesting direction for further study.

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
