# OpenReview forum: "Bandit Learning in Matching: Unknown Preferences On Both Sides"
_ICLR.cc/2024/Conference — Submitted to ICLR 2024_

### Official Review · Reviewer_V2Kp · 2023-10-30

**Soundness:** 3 good
**Presentation:** 4 excellent
**Contribution:** 2 fair
**Rating:** 5
**Confidence:** 3

**Summary:**

Existing works in matching bandits mainly focus on the one-sided setting where arms are aware of their own preferences. This paper considers a more general two-sided setting, i.e. participants on both sides have to learn their preferences over the other side through repeated interactions. The authors consider Round-Robin ETC in this two-sided setting and provided $log(T)$ regret analysis.

**Strengths:**

While a few typos are present, the paper is generally well-written and remarkably easy to comprehend. The authors have made significant efforts to simplify the presentation, using accessible language to explain complex algorithms and regret analysis.

**Weaknesses:**

1. Rationale for the Chosen Scenario:
In the context of the discussed setting, the true motivation remains somewhat elusive. Although this paper delves into a novel matching scenario where both parties' preferences are unknown, it lacks a concrete real example to substantiate the practical viability of the proposed algorithm.

Firstly, the proposed algorithm introduces an ETC-style approach, mandating an initial round-robin exploration. However, this random exploration may not align with the preferences of practical agents. Alternative exploration strategies, such as UCB or Thompson Sampling, might offer more practicality.

Secondly, the algorithm necessitates a communication phase, wherein the players will communicate through deliberate conflicts in an index-based order to communicate whether they have achieved confident estimations and the communication proceeds pairwise following the order of index. This represents a demanding requirement for real-world players.

Thirdly, the regret analysis assumes that every player adopts the proposed Algorithm 1 and arms adhere to $R$ sample-efficient strategies. Yet, in practical applications of two-sided matching, like those in marriage, college admissions, and labor markets, as outlined in the paper's introduction, meeting these conditions might not be feasible.

In essence, this paper lacks a demonstrable real-world application that would justify the prerequisites outlined in the algorithm and regret analysis.


2. Comparative Analysis with Pokharel and Das (2023):
Notably, Pokharel and Das (2023) also explored the same matching bandit scenario, where the preferences of both sides remain unknown. Although they did not provide a detailed regret analysis, it is imperative to numerically compare the proposed algorithm with their work. Surprisingly, in the current numerical studies outlined in Section D of the supplement, none of the existing algorithms were subjected to comparison.

Reference:
Gaurab Pokharel and Sanmay Das. "Converging to Stability in Two-Sided Bandits: The Case of Unknown Preferences on Both Sides of a Matching Market." arXiv preprint arXiv:2302.06176, 2023.

**Questions:**

see weakness. In the rebuttal stage, I would like to see more real justifications as well as numerical comparison with the literature.

---

> ### Author Response · Authors · 2023-11-18
>
> Thank you for your feedback. If you have any further concerns about the justifications and simulations, please let us know, and we are happy to provide further clarification.
>
> W1. We think it is rational for the players to adopt the algorithm we proposed, as it helps them match with their optimal stable pairs with low regret. We further specify the rationale for applying our algorithm from the following aspects.
>
>  (1) The choice of the ETC framework. Unlike traditional bandit problems, in matching bandits, players compete for arms and may encounter conflicts resulting in zero rewards. If players opt for UCB-type or TS-type strategies, they are likely to frequently encounter conflicts, making it difficult to obtain useful samples for learning their preferences and updating the UCB estimators. Consequently, converging to the optimal stable matching becomes more  challenging, leading to potentially much higher regret. For instance, in [Liu et. al, 2021], they illustrate an example where even the centralized UCB can not achieve sub-linear player optimal regret. Thus, it may not be rational for them to adopt such strategies. While random exploration may not perfectly align with the preferences of practical players, it serves as a useful and efficient method for both players and arms to gather sufficient samples for initially estimating their preferences and avoid conflicts.
>
> (2) The incentive in the communication process.  It is worth noting that it is not enough for a single player to learn her own preferences. The approach to player-optimal stable matching requires each participant to learn her own preferences and coordinate with each other. Communication is employed to facilitate this coordination and find a stable matching. Without the communication process, players may encounter conflicts, receive zero rewards, and fail to match with their optimal stable pairs. Therefore, the communication process actually helps the players match with their optimal stable arms effectively and with minimal cost.
>
> (3) The assumptions on strategies. We should emphasize that sample-efficient strategies include many different kinds of policies applicable in various real-world scenarios. Our assumption that arms adopt sample-efficient strategies is easily satisfied in reality. For instance, our assumption covers scenarios where some arms use the empirical mean estimator to choose which invitation to accept, while other players use the UCB estimator to select candidates.
>
> Regarding player-side strategies, it is a common assumption in the literature of decentralized bandit algorithm design that all players apply the same algorithm[Sankararaman et al., 2021][Liu et al., 2021][Kong and Li][Pokharel and Das, 2023][Zhang et al., 2022]. This assumption also captures the decentralized property, wherein all players lack global information before interactions. If we design different algorithms for different players, we would need to specify which set of players should apply which algorithms, and this would require global information or communication beforehand.
>
> W2. More simulations.
> Due to the time limit in the rebuttal period, we have added two cases of simulations to compare our algorithm's performance with the algorithm proposed in [Pokharel and Das, 2023], and we will incorporate more cases in our revision.  Figures and simulation details are shown in the supplementary materials. Moreover, several comments are provided below to compare the two algorithms.
>
> (1) Theoretical comments: The algorithm PCA-UCB is an extension of algorithm CA-UCB while CA-UCB only achieves a $O(\log ^2 T)$ regret bound compared with the player pessimal regret, even in the one-sided setting. Moreover, [Pokharel and Das 2023] make strong but not so realistic assumptions, such as the observation of the winning players (discussed in Section 2).
>
> (2) Simulation performance: From the figures, we can conclude that round-robin ETC outperforms PCA-UCB in both cases. Additionally, the results of round-robin ETC exhibit greater stability than those of PCA-UCB. The reason why PCA-UCB performs unstably and fails to obtain sublinear results  may be as follows:
>
> Firstly, in different runs of simulations, PCA-UCB may converge to different stable matchings instead of consistently converging to the player-optimal stable matching.
> This variability in convergence could be a significant challenge, as the player-optimal stable matching is more desirable for players. Furthermore, as we stated, in [Liu et al., 2021], they illustrate an example where even the centralized UCB cannot achieve sub-linear player-optimal regret.
>
> Secondly, when applying PCA-UCB, players adopt a UCB-type method to choose arms, resulting in insufficient samples for arms to learn their preferences. Consequently, arms may provide inaccurate feedback in the two-sided setting, potentially leading to unstable matching or an extended time to convergence.

---

> > ### Author Response · Authors · 2023-11-18
> > **References**
> >
> > [Pokharel and Das, 2023] Gaurab Pokharel and Sanmay Das. "Converging to Stability in Two-Sided Bandits: The Case of Unknown Preferences on Both Sides of a Matching Market." arXiv preprint arXiv:2302.06176, 2023.
> >
> > [Sankararaman et al., 2021] Sankararaman, Abishek, Soumya Basu, and Karthik Abinav Sankararaman. "Dominate or delete: Decentralized competing bandits in serial dictatorship." International Conference on Artificial Intelligence and Statistics. PMLR, 2021
> >
> > [Kong and Li, 2023] Fang Kong and Shuai Li. Player-optimal stable regret for bandit learning in matching markets. In Proceedings of the 2023 Annual ACM-SIAM Symposium on Discrete Algorithms (SODA), pp.1512–1522. SIAM, 2023
> >
> > [Liu et al., 2021] Lydia T Liu, Feng Ruan, Horia Mania, and Michael I Jordan. Bandit learning in decentralized
> > matching markets. Journal of Machine Learning Research, 22(211):1–34, 2021.
> >
> > [Zhang et al., 2022] Yirui Zhang, Siwei Wang, and Zhixuan Fang. Matching in multi-arm bandit with collision. Advances
> > in Neural Information Processing Systems, 35:9552–9563, 2022.

---

### Official Review · Reviewer_QoQ1 · 2023-10-31

**Soundness:** 3 good
**Presentation:** 2 fair
**Contribution:** 2 fair
**Rating:** 5
**Confidence:** 5

**Summary:**

This paper studies the bandit learning problem in matching markets where both sides of agents have unknown preferences. It proposes a round-robin ETC algorithm to let players and arms learn their preferences and find a partner in a stable matching. Theoretical upper bounds on the player-optimal and arm-pessimal regret bound are provided.

**Strengths:**

This paper considers the setting where both sides of market participants have unknown preferences for the first time. For this setting, the paper proposes a round-robin ETC algorithm and the associated communication protocol to let players and arms adaptively find their stable matching. The upper bound of stable regret for both the player side and arm side is provided.

**Weaknesses:**

1. The algorithmic design depends on the known D which reveals the relationship between the players' and arms' minimum preference gap. This assumption is unrealistic since both sides have unknown preferences.
2. The algorithm design is a direct extension of previous works (Kong and Li, 2023; Zhang et al., 2022). What is your main technical novelty?
3. Previous works assume N\le K as they consider the player-unknown setting to let each player have a chance to be matched. Why assume N\le K in the two-sided unknown setting?
4. In this two-sided setting, both sides of the participants are rational agents. The current approach though considers such a setting, the focus is still on the player side. For example, if we assume the player-side preferences to be known and arm-side preferences to be unknown, the optimal result should be on the arm-optimal stable matching. The paper lacks a discussion of the trade-off on the optimality of both sides.
5. Minor: The communication protocol is not clear enough. This part of writing needs to be polished.

**Questions:**

Please see above weakness.

---

> ### Author Response · Authors · 2023-11-15
>
> Thank you for your feedback. If you have any further concerns, please let us know, and we are happy to provide further clarification.
>
> W1. Known $D$. $D$ is merely an upper bound assumption and only reveals the comparative level between the arm side and player side, rather than providing a detailed value of the minimal gap. In real-world applications, players can estimate $D$ using available information. Moreover, we believe that preference learning is often easier for arms than for players ($D\le 1$), as arms typically have access to more information to help estimate their preferences. For example, companies (arms) often have access to workers' resumes to assist in estimating their preferences. Alternatively, players can simply assume a large value for $D$ when applying our algorithm.
>
> W2. Technical novelty.  The main similarity between our algorithm and the proposed algorithms in [Kong and Li, 2023][Zhang et al., 2022] lies in the ETC framework, which is also commonly employed in the design of bandit algorithms. Furthermore, we highlight several key differences from two aspects below, demonstrating our technical novelty in the meantime.
>
> (1) Different settings. The algorithms you mentioned all consider and apply solely to the one-sided setting, whereas ours is designed for the two-sided setting. In our work, we introduce and further specify the two-sided model, considering rational strategies from the arm side. Conditional on the newly proposed setting, we introduce our algorithm to study convergence to stable matching. Additionally, in [Kong and Li, 2023], there is an assumption of observing all winning players, which is a strong but impractical assumption (discussed in Section 2).
>
> (2)Algorithm Designs and Theoretical Analysis. The algorithms in [Kong and Li, 2023][Zhang et al., 2022] focus solely on the exploration and exploitation tradeoff for the player side, while our designed algorithm needs to coordinate players against the unreliable feedback from the arm side simultaneously. The theoretical analysis methods also differ. For instance, when considering the two-sided setting, a new method for analyzing the correctness of arms' feedback is necessary. In our paper, we propose lemmas to demonstrate that as long as players have confidence in the estimations of arm utilities, the arms will provide precise feedback with high probability. Different analyses for the player side, such as the analysis of deliberate conflict, are also presented in our paper.
>
> Lastly, we have summarized the relaxed assumptions compared with [Kong and Li, 2023] and the improved regret bound compared with [Zhang et al., 2022] in Table 1.
>
> W3. The assumption of $N\le K$. Since our paper focuses on the extended setting of two-sided unknown preferences in matching bandits, we adhere to the commonly used assumption of $N \leq K$ in matching bandit literature. Additionally, as you mentioned, previous works assume $N \leq K$ when considering the player-unknown setting to ensure each player has a chance to be matched. In the context of the two-sided unknown setting, the corresponding assumption of $N=K$ is more natural to provide every participant with an opportunity to be matched. Our assumption of $N \leq K$ encompasses this condition.
>
> W4. Rational on the both sides. The reason why we focus on the player-optimal stable matching is as follows: In the two-sided setting, players proactively propose to arms, and arms can only decide whether to accept the players. From our perspective in this scenario, the arms play a role analogous to the agents being proposed to in the Gale-Shapley algorithm, while players serve as the proposer side. The rationale behind considering player-optimal stable matching is to explore whether similar results to the Gale-Shapley algorithm (specifically, the convergence to the optimal stable matching for the proposing side) are applicable in the two-sided unknown setting. We believe that optimality may be influenced more by the proposing dynamics than by the unknown preferences. Furthermore, in our paper, the definition of our proposed sample-efficient strategies encompasses various rational strategies without imposing many restrictions. We have also mentioned the potential strategy competition for optimality between the two sides as a noteworthy direction for future research.
>
> W5. Presentations. we have already polish our presentation in the revision.
>
> [Kong and Li, 2023] Fang Kong and Shuai Li. Player-optimal stable regret for bandit learning in matching markets. In Proceedings of the 2023 Annual ACM-SIAM Symposium on Discrete Algorithms (SODA), pp.1512–1522. SIAM, 2023
> [Zhang et al., 2022] Yirui Zhang, Siwei Wang, and Zhixuan Fang. Matching in multi-arm bandit with collision. Advances
> in Neural Information Processing Systems, 35:9552–9563, 2022.

---

### Official Review · Reviewer_hbNf · 2023-11-01

**Soundness:** 3 good
**Presentation:** 2 fair
**Contribution:** 3 good
**Rating:** 5
**Confidence:** 3

**Summary:**

The paper considers a 2-sided matching problem with N players and K $\geq$ N arms. No agent on either side is aware of their true preferences a priori; these must be leant sequentially from noisy bandit feedback over T rounds of interaction between the 2 sides. In any given round, the players simultaneously propose to arms of their choice based on some decision rule. If multiple players propose to the same arm, the arm chooses a player of her choice based on her decision rule. The authors propose a solution concept whereby players follow the same decision rule (dubbed Round robin ETC) and each arm follows some "sample efficient" learning strategy. Under this premise, the player-optimal stable regret is shown to grow logarithmically in T and so is the arm-pessimal stable regret.

**Strengths:**

I think this paper contributes well to the line of work on two-sided matching via bandit learning. In particular, theoretical results for the setting where both sides learn over time are new. Additionally, positive results are shown wrt a stronger benchmark (player-optimal matching), which is novel. Other than that, several assumptions common in extant literature (e.g., conditions pertaining to a unique stable match or globally best arms, etc.) seem to have been relaxed.

**Weaknesses:**

My concern is there are too many moving parts in the algorithm (the stated version is a composition of several modules) and its presentation, in the current form, is not ideal. The paper certainly could benefit from a better exposition of Section 3.2 (as would the reader).

**Questions:**

The upper bounds in Theorem 1 depend on R implicitly through c. Please define c in the statement of the theorem.

---

> ### Author Response · Authors · 2023-11-15
>
> Thank you for your feedback. If you have further concerns about the presentation, please let us know, and we would be delighted to incorporate your additional advice to help us polish our presentation in our revision.
>
> Weaknesses:
>  The structure and the presentation: We have already reorganized the algorithm modules and polished the writing in the revision.
>
> Questions:
> Thanks for pointing it out, and we have already added it in our revision. We also emphasize that $R$ is usually a constant no larger than $16$, given that commonly used methods such as UCB and empirical mean are both sample-efficient strategies that satisfy $R=16$.

---

### Official Review · Reviewer_X1qd · 2023-11-07

**Soundness:** 3 good
**Presentation:** 2 fair
**Contribution:** 2 fair
**Rating:** 3
**Confidence:** 3

**Summary:**

This paper studies the problem of two-sided matching bandits in the style of Das and Kamenica (IJCAI 2005), where both the players and arms have unknown preferences over one another and the market is decentralized. The goal is for the players to learn their optimal stable match with low regret. The key challenges are that (1) both sides need to efficiently learn their preferences through limited samples, (2) arms adopt sample-efficient strategies in choosing players, leading to unreliable feedback, and (3) lack of communication and coordination among decentralized players. In terms of results, the authors show that learning can be achieved with $O(\log T)$ instance-dependent player-optimal regret via a round-robin explore-then-commit approach.

**Strengths:**

- Develops a new model of decentralized two-sided matching, extending that of Das and Kamenica and Liu et al. Particularly interesting is the authors’ modeling of arms’ learning processes via “sample-efficient” algorithms.
- Provides a thorough analysis of their model, including an optimal $O(\log T)$ instance-dependent learning algorithm for decentralized agents.

**Weaknesses:**

- It is inaccurate to say that there are no other works on matching bandits where both sides’ preferences are unknown. For instance, the paper of Jagadeesan et al. (JACM, 2023) considers a model of matching bandits on a centralized platform where all agents do not know their preferences to start with.
- I am not fully convinced by the motivation/modeling assumptions here. It would be helpful if there discussion of a specific empirical setting this paper sought to model. (In particular, I do not find the example given of online marketplaces particularly compelling, as my understanding is that most such platforms run based on *centralized* recommendations.)
- What are the insights that should be taken away from this work? Is it new qualitative understanding, or is it the results themselves? I believe the paper would benefit from a more thorough discussion of takeaways/insights beyond the theorem statements.

**Questions:**

- What are the empirical settings that motivate this particular model of matching bandits?
- What are the key qualitative insights that the authors would like to emphasize?

---

> ### Author Response · Authors · 2023-11-15
>
> Thank you for your feedback. If you have further concerns about the model and the insights, please let us know, and we would be delighted to incorporate your additional advice to help us add a discussion part in our revision.
>
> W1. Related works. Thank you for pointing it out, and we have already corrected our presentation in our revision. However, we want to emphasize that our paper considers the two-sided setting from a substantially different perspective than the work of Jagadeesan et al. (JACM, 2023), and works addressing two-sided unknown preferences in matching bandits remain scarce.
>
> Jagadeesan et al. investigate matching markets using the stochastic contextual bandit model, where, at each round, the platform selects a market outcome with the goal of minimizing cumulative instability. In brief, Jagadeesan et al. focus on the strategy of the central platform, which can directly determine the matching outcomes.
>
> In our setting, we consider the strategy of the participants. Every participant on both sides acts individually, and the collective actions of all participants jointly determine the matching result. Minor differences, such as the existence of transfers, also vary.
>
> W2&Q1. Model: A practical case as mentioned in the paper of [Sankararaman et al. 2021] is that of scheduling jobs
> to servers in an online marketplace, e.g., crowd sourcing platforms (Upwork and TaskRabbit), question answering platforms (Quora, Stack Overflow)[Shah et al., 2020].  In these marketplaces, rewards are often stochastic and can only be obtained upon completion. Furthermore, given the frequent and repeated interactions, these platforms aim to optimize cumulative rewards over an extended time horizon, solving thousands of instances every hour.
>
> It is true that those platforms may provide recommendations. However, the workers still act individually and independently, capturing the decentralized setting; that is, there is no central communicator that enable players to coordinate their actions directly with each other. Furthermore, due to privacy concerns, workers may choose not to disclose their received rewards to the centralized platform [Rees-Jones and Skowronek, 2018]. For scalability concern, decentralized solutions are also preferred [Larsson, 2018].  Indeed, most of the related work mentioned in our paper addresses matching bandits from a decentralized perspective.
>
> W3&Q2. Insight: Based on classical results in the matching market, the optimal matching for the proposing side can be obtained using the Gale-Shapley algorithm. The insight arises from our curiosity about whether this result can still hold in a two-sided unknown setting. By designing a specific algorithm and providing rigorous analysis, we find out that the goal is attainable, i.e. the market converges to that optimal stable matching at
> a logarithmic rate. Furthermore, the algorithm design and theoretical analysis methods themselves may also serve as a preliminary step for future studies in the two-sided matching bandits.
>
>
> [Sankararaman et al. 2021] Sankararaman, Abishek, Soumya Basu, and Karthik Abinav Sankararaman. "Dominate or delete: Decentralized competing bandits in serial dictatorship." International Conference on Artificial Intelligence and Statistics. PMLR, 2021.
>
> [Shah et al., 2020] Shah, V., Gulikers, L., Massouli´e,
> L., and Vojnovi´c, M. (2020). Adaptive matching for
> expert systems with uncertain task types. Opera-
> tions Research.
>
> [Larsson, 2018] Larsson, S. (2018). Law, society and
> digital platforms: Normative aspects of large-scale
> data-driven tech companies. In The RCSL-SDJ Lis-
> bon Meeting 2018” Law and Citizenship Beyond The
> States”.
>
> [Rees-Jones and Skowronek, 2018] Rees-Jones, A.
> and Skowronek, S. (2018). An experimental
> investigation of preference misrepresentation in
> the residency match. Proceedings of the National
> Academy of Sciences, 115(45):11471–11476

---

### Meta-Review · Area_Chair_3EC2 · 2023-12-04

**Metareview:**

This paper looks at a bandit problem where a matching between arms and agents is output at each time step. The key thing is that both arms and agents are unaware of their preferences hence they are learned on the fly, with standard bandit techniques.

This paper seems correct to me but not really exciting. It does not bring anything particularly new to the literature but is rather a nice (and not necessarily trivial) application of existing techniques (ETC algorithms, combined with upper/lower estimate bounds).

Therefore, I find it too incremental to reach the selection bar of ICML, and must recommend rejection

**Justification For Why Not Higher Score:**

It does not reach the bar

**Justification For Why Not Lower Score:**

N/A

---

### Decision · Program_Chairs · 2024-01-16

Reject